FERMILAB-PUB-21-536-T, MCNET-21-30

# Reducing negative weights in Monte Carlo event generation with Sherpa

K. Danziger[1], S. Höche[2], F. Siegert[1*],

**1** Institut für Kern- und Teilchenphysik, TU Dresden, Dresden, Germany
**2** Fermi National Accelerator Laboratory, Batavia, USA
* frank.siegert@cern.ch

October 29, 2021

## Abstract

An increase in theoretical precision of Monte Carlo event generators is typically accompanied by an increased need for computational resources. One major obstacle are negative weighted events, which appear in Monte Carlo simulations with higher perturbative accuracy. While they can be handled somewhat easily in fixed-order calculations, they are a major concern for particle level event simulations. In this article, the origin of negative weights in the S-Mc@Nlo method is reviewed and mechanisms to reduce the negative weight fraction in simulations with the Sherpa event generator are presented, with a focus on $V$+jets and $t\bar{t}$+jets simulations.

# 1 Introduction

Computing challenges and limitations have a large impact on the feasibility and ambition of the physics program at the Large Hadron Collider (LHC). A significant part of the computing needs of the major LHC experiments arises from the simulation of signal and background events and their interactions with the detectors [1]. With more luminosity being delivered to the experiments in future LHC data taking periods, the number of events that can be simulated with the current resources and computing model will likely not match the number of events recorded in experimental data [2].

The simulation of physics processes relies heavily on general-purpose Monte Carlo event generators like HERWIG7 [3], PYTHIA8 [4], or SHERPA [5]. They provide a realistic simulation of the final state of proton-proton collisions including effects that go beyond what is possible in fixed-order calculations, like the resummation of soft or collinear QCD radiation in a parton shower model, the fragmentation into primordial hadrons and their decays, and the possibility of multiple partonic interactions within the same collision.

In the last decade, these generators have been extended to include a more accurate calculation of the hard scattering process within the framework of perturbative QCD. The matching of next-to-leading order (NLO) matrix elements to parton showers (PS) [6–9] and the subsequent merging of NLO+PS simulations for several jet multiplicities into an inclusive sample [10–14] results in simulations, which achieve the highest precision for many of the processes critical for the ongoing experimental analyses. At the same time, these simulations come with practical disadvantages. Not only are the calculations much more complex and thus require more computational resources per event, but they also often contain events which are not distributed naturally with unit weights ("unweighted") but instead feature negative weights that represent subtraction terms in the calculation. These events reduce the statistical power of the event sample, necessitating the simulation of larger samples using correspondingly larger computational resources.

The problem has received much attention recently [15], and various solutions have been proposed. Resampling methods [16–18] for example can be used to redistribute the event weights a posteriori in a largely observable independent manner. Generator or algorithm specific techniques such as the MC@NLO-$\Delta$ method [19] can be used to eliminate a large fraction of negative weighted events a priori. In this article we present various options that are specific to S-MC@NLO and MEPS@NLO simulations with the SHERPA event generator. After a brief introduction to the various sources of negative weights we present the mitigation techniques and discuss the impact of their implementation not only on the negative weight fraction but also on the predictive power of the event sample. We focus on the two most relevant background processes for LHC physics, $V$+jets and $t\bar{t}$+jets production.

## 2 Impact of negative event weights

Naively, each event in a simulation sample is associated to an event weight, which corresponds to the differential cross section of the phase space point the event represents. For experimental applications it is particularly interesting to obtain event samples with equal weights, as then the events represent a natural distribution of the simulated process like it would be found in collision events. Thus, most event generators implement an unweighting procedure to reject events in such a way, that the output sample follows a natural distribution with unit weights. As will be shown in Sec. 3, events can also be associated with negative weights in some cases where a partial differential cross section becomes negative. The unweighting procedure in such cases will reject positively and negatively weighted events separately, such that each set is unweighted to weights $w = \pm 1$, respectively.

Samples containing negatively weighted events have a reduced statistical power. Considering a negative weight fraction $\varepsilon$ and, up to a sign, equally weighted events with weights $w = \pm c$, where $c$ is a constant larger than zero, the statistical accuracy of an event sample of size $N$ can be evaluated as

$$\frac{\sqrt{\sum_i w_i^2}}{\sum_i w_i} = \frac{\sqrt{c^2 \cdot N}}{c \cdot (1 - 2\varepsilon) \cdot N} = \frac{1}{(1 - 2\varepsilon)} \cdot \frac{1}{\sqrt{N}} = \frac{1}{\sqrt{N_{eff}}}. \tag{1}$$

This yields the effective number of events, given by $N_{eff} = (1 - 2\varepsilon)^2 \cdot N$, which incorporates the diminished statistics in the presence of negatively weighted events. Hence, one needs to generate a factor

$$f(\varepsilon) = \frac{1}{(1 - 2\varepsilon)^2} \tag{2}$$

more events to get the same statistical accuracy in comparison to only positively weighted events. Figure 1 illustrates this factor $f$ as a function of the negative weight fraction $\varepsilon$. As can be seen, this factor increases dramatically with $\varepsilon$. For instance, if a quarter of the events have a negative weight, a four times larger sample has to be produced, and if the negative weight fraction rises to $\varepsilon = 40\%$ one needs to generate 25 times more events. While this raises the computational cost already on the event generation side, the more pressing issue is the subsequent detector simulation, making up the major part in terms of CPU consumption. If the negative weight fraction is too large, using full detector simulations becomes impractical. Thus, it is of utmost importance to eliminate negative weights whenever possible.

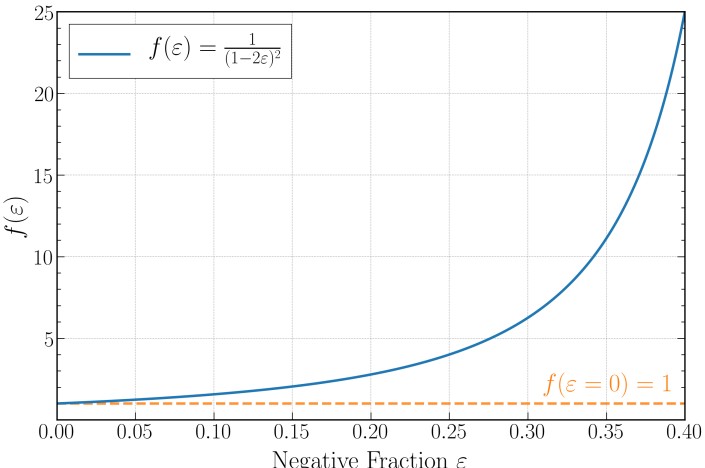

**Figure 1:** *Dependence of the factor $f$, quantifying the excess of events to be generated for the same statistical accuracy as in the case of solely positive weighted events, on the negative weight fraction $\varepsilon$.*

## 3 Sources of negative weights in S-Mc@Nlo and Meps@Nlo

Negatively weighted events are commonplace when going beyond the leading order in perturbation theory for non-trivial processes. To illustrate this, we give a brief summary of their origin in the S-Mc@Nlo matching scheme implemented in Sherpa.

Numerical NLO calculations are typically performed using infrared (IR) subtraction techniques [20–22]. These methods isolate the IR divergences in the real-emission corrections and virtual corrections, and allow to define integrands that are finite in four dimensions for any IR safe observable, and are thus integrable with Monte Carlo methods. In the dipole method [21, 22], soft singularities of the real-emission matrix elements are mapped onto collinear sectors and isolated into a sum of dipole terms $D^{(\mathcal{S})}$, which can be used to render the real emission matrix elements $R$ finite. The corresponding integrated subtraction terms $I^{(\mathcal{S})}$ in turn render the virtual corrections finite, such that the differential dipole-subtracted cross-section at NLO can schematically be written as

$$\mathrm{d}\sigma^{\mathrm{NLO}} = \mathrm{d}\Phi_B \Big[ B(\Phi_B) + \tilde{V}(\Phi_B) + I^{(\mathcal{S})}(\Phi_B) \Big] + \mathrm{d}\Phi_R \Big[ R(\Phi_R) - D^{(\mathcal{S})}(\Phi_R) \Big], \qquad (3)$$

Here $B$ and $\tilde{V}$ are the Born and UV renormalized virtual contribution, while $\mathrm{d}\Phi_B$ and $\mathrm{d}\Phi_R$ denote the phase-space element for the Born and real-emission configuration, respectively. The dipole terms are determined by means of insertion operators that correlate the color- and spin-dependent Born amplitudes according to the structure of QCD in the soft-gluon limit and the collinear limit. Schematically we have

$$D^{(\mathcal{S})}(\Phi_R) \rightarrow \sum_{ijk} \langle M_B(\Phi_B) | \frac{\mathbf{T}_{ij}\mathbf{T}_k}{\mathbf{T}_i^2} V_{ijk}(\Phi_R, \Phi_B) | M_B(\Phi_B) \rangle \,, \qquad (4)$$

where the sum runs over all parton indices, $\mathbf{T}_i$ is the color insertion operator for a parton $i$ [23], and the spin-dependent splitting kernels $V_{ijk}$ are given in [21]. We note that, depending on the precise value of the color correlator $\mathbf{T}_{ij}\mathbf{T}_k$, individual dipole terms can be either positive or negative valued.

Born-like events, corresponding to the first square bracket in Eq. (3), and real-emission like events, corresponding to the second square bracket in Eq. (3), are generated separately in practice. Negative event weights therefore occur in the calculation if the dipole approximation overestimates the real-emission matrix element.

This situation is complicated additionally by the matching to a parton shower resummation. In the Mc@Nlo matching method, an additional subtraction term is introduced, which removes the parton shower approximation from the real-emission like events and adds it in integrated form to the Born-like events. This modified subtraction technique removes the overlap between the parton-shower approximation and the fixed-order NLO result. Denoting the corresponding MC subtraction terms by $D^{(\mathcal{A})}$, Eq. (3) becomes

$$
\begin{aligned}
\mathrm{d}\sigma^{\mathrm{NLO}} = \mathrm{d}\Phi_B\Big[B(\Phi_B) + \tilde{V}(\Phi_B) + I^{(\mathcal{S})}(\Phi_B)\Big] + \mathrm{d}\Phi_R\Big[D^{(\mathcal{A})}(\Phi_R) - D^{(\mathcal{S})}(\Phi_R)\Big] \\
+ \mathrm{d}\Phi_R\Big[R(\Phi_R) - D^{(\mathcal{A})}(\Phi_R)\Big].
\end{aligned}
\tag{5}
$$

Standard parton showers are based on the leading color and leading spin approximation, and do therefore not capture the complete singularity structure of Eq. (4). The corresponding parton-shower splitting kernels would not suffice to define a locally IR finite differential cross section in Eq. (3). In the S-Mc@Nlo prescription, $D^{(\mathcal{A})}$ is therefore defined with the help of the dipole terms $D^{(\mathcal{S})}$, and set to zero outside the resummation region [24]:

$$
D^{(\mathcal{A})} = D^{(\mathcal{S})}\Theta(\mu_Q^2 - t).
\tag{6}
$$

Applying the parton shower resummation using $D^{(\mathcal{A})}$ as splitting kernels yields the matching formula

$$
\begin{aligned}
\sigma_{\text{S-Mc@Nlo}} = \int \mathrm{d}\Phi_B \bar{B}^{(\mathcal{A})}(\Phi_B)\Bigg[\underbrace{\bar{\Delta}^{(\mathcal{A})}(t_c, \mu_Q^2)}_{\text{unresolved}} + \underbrace{\int_{t_c}^{\mu_Q^2} \mathrm{d}\Phi_1 \frac{D^{(\mathcal{A})}(\Phi_B, \Phi_1)}{B(\Phi_B)}\ \bar{\Delta}^{(\mathcal{A})}(t, \mu_Q^2)}_{\text{resolved, singular}}\Bigg] \\
+ \int \mathrm{d}\Phi_R\ \underbrace{\Big[R(\Phi_R) - D^{(\mathcal{A})}(\Phi_R)\Big]}_{\text{resolved, non-singular,}\ \equiv\ H^{(\mathcal{A})}},
\end{aligned}
\tag{7}
$$

where $\bar{B}^{(\mathcal{A})}$ denotes the so-called next-to-leading order weighted Born cross-section given by

$$
\bar{B}^{(\mathcal{A})}(\Phi_B) = B(\Phi_B) + \tilde{V}(\Phi_B) + I^{(\mathcal{S})}(\Phi_B) + \int \mathrm{d}\Phi_1\Big[D^{(\mathcal{A})}(\Phi_B, \Phi_1) - D^{(\mathcal{S})}(\Phi_B, \Phi_1)\Big]
\tag{8}
$$

and $H^{(\mathcal{A})} = R - D^{(\mathcal{A})}$ describes the hard remainder term. The first emission is described in terms of a modified Sudakov form factor

$$
\bar{\Delta}^{(\mathcal{A})}(t, t') = \exp\Bigg\{-\int_t^{t'} \mathrm{d}\Phi_1 \frac{D^{(\mathcal{A})}(\Phi_B, \Phi_1)}{B(\Phi_B)}\Bigg\}.
\tag{9}
$$

The first line in Eq. (7) corresponds to so-called $\mathbb{S}$-events, where the first emission is generated as in the standard parton shower, but including the full colour structure. In contrast, the second line refers to $\mathbb{H}$-events, where a hard emission is generated pursuant to the subtracted real-emission matrix element and the parton shower generates subsequent emissions only. It

is apparent that the differential cross section weight of $\mathbb{H}$-events can become negative due to the modified subtraction procedure. Furthermore, also $\mathbb{S}$-events can induce negative weights due to the exponentiation of negative valued subtraction terms corresponding to sub-leading colour configurations in the approximated real emission matrix elements of Eq. (4).

The MEPS@NLO merging procedure [10, 11] is an extension of both the S-MC@NLO matching method and the CKKW merging algorithm, which allows to combine matched NLO calculations for varying jet multiplicity by an appropriate overlap removal through Sudakov reweighting and explicit subtraction of NLO fixed-order corrections already accounted for by the parton shower. As such, when applied to NLO fixed-order inputs, the MEPS@NLO algorithm has the same sources of negative weights as the S-MC@NLO matching method described above. In addition, when applied to LO inputs, the MEPS@NLO algorithm generates a Born-local $K$-factor that enables the smooth merging between the matrix elements of varying jet multiplicity [10, 11] and is a further source of negative weighted events, because it is evaluated in a Monte-Carlo fashion.

# 4 Mechanisms to improve the negative weight fraction

This section will discuss three mechanisms to reduce the negative weight fraction within SHERPA S-MC@NLO and MEPS@NLO events. It is worth noting, that all improvements relevant for S-MC@NLO simulations will also benefit MEPS@NLO simulations, since the former are components of the latter.

To define effective approaches for the reduction of negative weights, it is instructive to separate their two sources in S-MC@NLO events as described in Sec. 3. This is straightforward, because these events can appear as $\mathbb{S}$ and $\mathbb{H}$ events, which correspond to negative weights from subleading colour configurations and from the subtraction procedure, respectively. As can be seen in Table 1, $\mathbb{S}$-events take up a major share of the negatively weighted events for typical $Z$+jets and $t\bar{t}$+jets setups[1].

| | Positive Fraction | Negative Fraction |
|---|---|---|
| $\mathbb{S} + \mathbb{H}$ | 82% | 18% |
| thereof $\mathbb{S}$ | 88% | 58% |
| thereof $\mathbb{H}$ | 12% | 42% |

**(a)** $pp \rightarrow$ $Z$+0,1,2 jets@LO+3 jets@NLO

| | Positive Fraction | Negative Fraction |
|---|---|---|
| $\mathbb{S} + \mathbb{H}$ | 75% | 25% |
| thereof $\mathbb{S}$ | 91% | 72% |
| thereof $\mathbb{H}$ | 9% | 28% |

**(b)** $pp \rightarrow$ $t\bar{t}$+0,1 jets@LO+2,3 jets@NLO

**Table 1:** *Percentage of positive and negative weighted events and their composition of $\mathbb{S}$ and $\mathbb{H}$-events for proton-proton collisions at a centre-of-mass energy of 13 TeV.*

## 4.1 Leading Colour Approximation

One possibility to reduce the negative weight fraction is to establish and validate a leading colour and leading spin approximation in the matching of NLO calculations to the parton

---

[1]The exact setup details will be defined in Sec. 5.

shower. This approximation is motivated by the fact that corrections to the large $N_C$-limit in processes with two color charged particles at the leading order are typically suppressed by $1/N_C^2$, which would lead to corrections of the order of 10%. This suppression is particularly relevant in the context of the standard candles $pp \to W/Z$ at the LHC, and in the context of Higgs-boson production through gluon fusion and the associated production $pp \to HW^{\pm}$ and $pp \to HZ$, as well as the corresponding Standard Model backgrounds.

Using the S-Mc@Nlo algorithm, as described in Section 3, in a leading colour, leading spin mode amounts to dropping the analytic weights introduced in connection with the colour correct one-step parton shower [8], and restricting the emitting color dipoles to the ones that are present in the large-$N_c$ limit. This requires a projection of the hard matrix element onto the possible partial amplitudes, and selection of a configuration according to their magnitude squared. The precise details are given in [25, 26].

We note that the elimination of spin and color correlations from the S-Mc@Nlo procedure can be motivated theoretically by the fact that the parton shower is activate in the region of phase space which is dominated by large logarithms. The correct description of spin and color effects in this region would in principle require to include them in the parton-shower simulation at all orders. The quality of the matched simulation should therefore not be affected significantly when a single-emission effect is altered. However, the actual phase space covered by the parton shower does typically extend far into the non-logarithmic region (e.g. $p_{T,V} \sim m_V$ in $pp \to V$), and therefore the effects included through S-Mc@Nlo may play a non-negligible role. While the structure of color singlet production at hadron colliders effectively excludes any significant effect due to the factorization of the color matrix (see for example [21]), more complex Born processes may require a different treatment [27]. We therefore recommend the careful validation of the leading colour, leading spin approximation on a process-by-process and observable-by-observable basis. In this context it is important to have the complete S-Mc@Nlo algorithm available.

## 4.2   ℍ-Event Shower Interplay

The ℍ-events in S-Mc@Nlo matched simulations are identified as hard corrections to 𝕊-events, and they do not exhibit any singularities in the single soft or double collinear regions of the phase space. In a multi-jet merged approach, it is therefore necessary to apply a Sudakov reweighting to an $n$-jet ℍ-event between the scales of the zeroth and the $n$-th jet, but a reweighting between the scale of the $n$-th jet and lowest clustering scale is not strictly required. If applied, such a reweighting corresponds to treating the ℍ-events like leading-order contributions to a multi-jet merged prediction, rather than finite remainders [10]. The additional Sudakov suppression reduces their cross section in the soft region of phase space, and consequently reduces the negative weight fraction. It neither influences the formal accuracy of the matched result nor the negative weight fraction in the high-$p_T$ region. However, it is preferred in order to achieve a continuous transition to higher-multiplicity Born-level predictions. The implementation of the reweighting has also been discussed in the context of Mc@Nlo matching at fixed multiplicity [19].

## 4.3   Core local $K$-factor

The negative weight fraction at next-to-leading order is increasing when going to higher multiplicities in the final-state due to the higher complexity of the processes. This fact can be

exploited to reduce the negative weight fraction by reducing the cases where high-multiplicity NLO matrix elements are evaluated without being phenomenologically relevant.

In the context of multijet merging employing the MEPS@NLO method, a differential factor $k_n^{(\mathcal{A})}$ is introduced for the purpose of a smooth merging between the matrix elements of varying jet multiplicity at NLO and LO [10,11]. This K-factor is originally applied for high-multiplicity LO matrix element events (e.g. $V+4$ jets) by clustering them with a reverse shower algorithm until they correspond to a multiplicity for which an NLO matrix element exists (e.g. $V+2$ jets, corresponding to $k_{n=2}^{(\mathcal{A})}$).

One can instead determine $K^{(\mathcal{A})}$ from the lowest underlying core process, e.g. $V+0$ jets ($k_{n=0}^{(\mathcal{A})}$). Given the smaller negative weight fraction of this low-multiplicity process, this is expected to further reduce $\varepsilon$ in higher transverse momentum regions where the high-multiplicity LO matrix elements contribute most.

Electroweak virtual corrections can induce significant effects in the high transverse momentum region of $V+$jets final states and similar processes. These corrections originate in the emergence of electroweak Sudakov logarithms [28], which have recently been implemented in SHERPA [29]. They can also be modeled very efficiently in the $EW_{virt}$ approximation [30]. Defining local $K$-factors based on the lowest multiplicity final state has the drawback that such corrections cannot be applied. Consequently, using $k_{n=0}^{(\mathcal{A})}$ may become undesirable if electroweak corrections are relevant, and the applicability of the approximation needs to be assessed depending on the process and observable under consideration.

## 5 Results

This section presents results obtained for the previously introduced mechanisms with the aim of reducing the amount of negatively weighted events. Two process setups are investigated, namely

$$pp \to Z(\to e^-e^+)+0,1,2 \text{ jets@NLO}+3 \text{ jets@LO}$$

and

$$pp \to t\bar{t}+0,1 \text{ jets@NLO}+2,3 \text{ jets@LO},$$

where the notation specifies, which jet multiplicity processes are simulated at NLO accuracy and which ones are simulated at LO accuracy within the sample merged according to the MEPS@NLO prescription [10]. Both processes are studied at a centre-of-mass energy of $\sqrt{13}$ TeV using the NNPDF 3.0 NNLO PDF set with $\alpha_S = 0.118$. The merging cut $Q_{cut}$ is set to 20 GeV in the case of $Z$ production and to 30 GeV for $t\bar{t}+$jets. All results are obtained using version 2.2.8 of SHERPA with virtual contributions provided by OPENLOOPS version 1.3.1 [31]. In the following, all displayed uncertainties correspond to the statistical Monte Carlo uncertainty.

Analyses of the physics performance and the negative weight fractions are performed using the Rivet framework [32]. In the case of $Z$ production, an invariant mass of $66 \text{ GeV} < m_{e^-e^+} < 116 \text{ GeV}$ is required in the analysis. Furthermore, jets are defined using the anti-$k_t$ clustering algorithm [33] with $R = 0.4$ and $p_T > 20$ GeV.

## 5.1   $Z$+jets

The study of vector boson production in association with jets plays an important role in the ongoing physics program of the LHC, where they can be measured with high precision and thus enable stringent tests of the Standard Model. Furthermore, these processes contribute as background to some of the most important searches for physics beyond the Standard Model, e.g. dark matter or supersymmetric signatures. As a result, vector boson plus jet processes constitute a vast amount of the Monte Carlo samples produced for the ATLAS and CMS experiments at the LHC, making the reduction of negatively weighted events for these processes a priority.

|  | Negative Weight Fraction |
| :---: | :---: |
| Default | 18.1% |
| Leading Colour Mode | 14.0% |
| + shower veto on $\mathbb{H}$-events | 9.6% |
| + local K-factor from core | 9.1% |

**Table 2:**  *Evolution of the mean negative weight fraction for pp $\rightarrow$ Z+0,1,2 jets@LO +3 jets@NLO.*

Table 2 shows the evolution of the negative weight fraction $\varepsilon$ for $Z$+jets using the approaches introduced in the previous section. As can be seen, $\varepsilon$ is reduced by roughly a factor of two, from 18.1% down to 9.1%. A reduction of negative weights is only useful if the precision of the physics prediction is not altered. Figures 2 and 3 show a comparison of the discussed approaches for various Monte Carlo validation observables. There, the lower two ratio plots display the negative fraction $\varepsilon$ and the corresponding factor $f(\varepsilon)$, which indicates how many more events are needed on average in order to achieve the same statistical accuracy as compared to only positively weighted events in each bin.

Let us first review the physics performance of the various simplifications. Various levels of agreement between the nominal and negative-weight-reduced predictions can be observed. The leading-colour approximation performs very well and reproduces the nominal prediction at the %-level, certainly much better than the naive expectation of a 10% accuracy. The additional Sudakov veto on $\mathbb{H}$-events causes a rate reduction of roughly 5% at intermediate values of $p_\perp^Z$ but otherwise the prediction is not significantly modified. The core local $k$-factor does not show a strong effect, implying a high level of universality between the $Z + n$-jet NLO predictions. In summary, the observed deviations in all methods are small compared to the systematic uncertainties that have to be taken into account in such predictions, e.g. from perturbative scale variations or ambiguities in the shower algorithm.

The leading colour mode reduces the number of negative weights in the low $p_T$ region, while it leaves $\varepsilon$ unchanged for large transverse momenta. The additional Sudakov suppression on $\mathbb{H}$-events further decreases the negative fraction for low to intermediate $p_T$, in the region where $\mathbb{H}$-events already contribute and resummation effects are still relevant. Defining the local K-factor on the core process lowers $\varepsilon$ mainly for large transverse momenta, as expected in this region, where multi-leg processes are dominant and taken into account through LO matrix elements.

It is essential for Monte Carlo event generators to describe the data as good as possible. Measurements of the production of a $Z$ boson in association with jets in proton-proton collisions

at a centre-of-mass energy of 13 TeV using 3.16 fb$^{-1}$ of data, collected by the ATLAS experiment at the LHC [34], are compared to SHERPA in Figure 4. The Monte Carlo predictions are in good agreement with the measurement, while again reducing the negative weight fraction significantly by employing the above discussed methods.

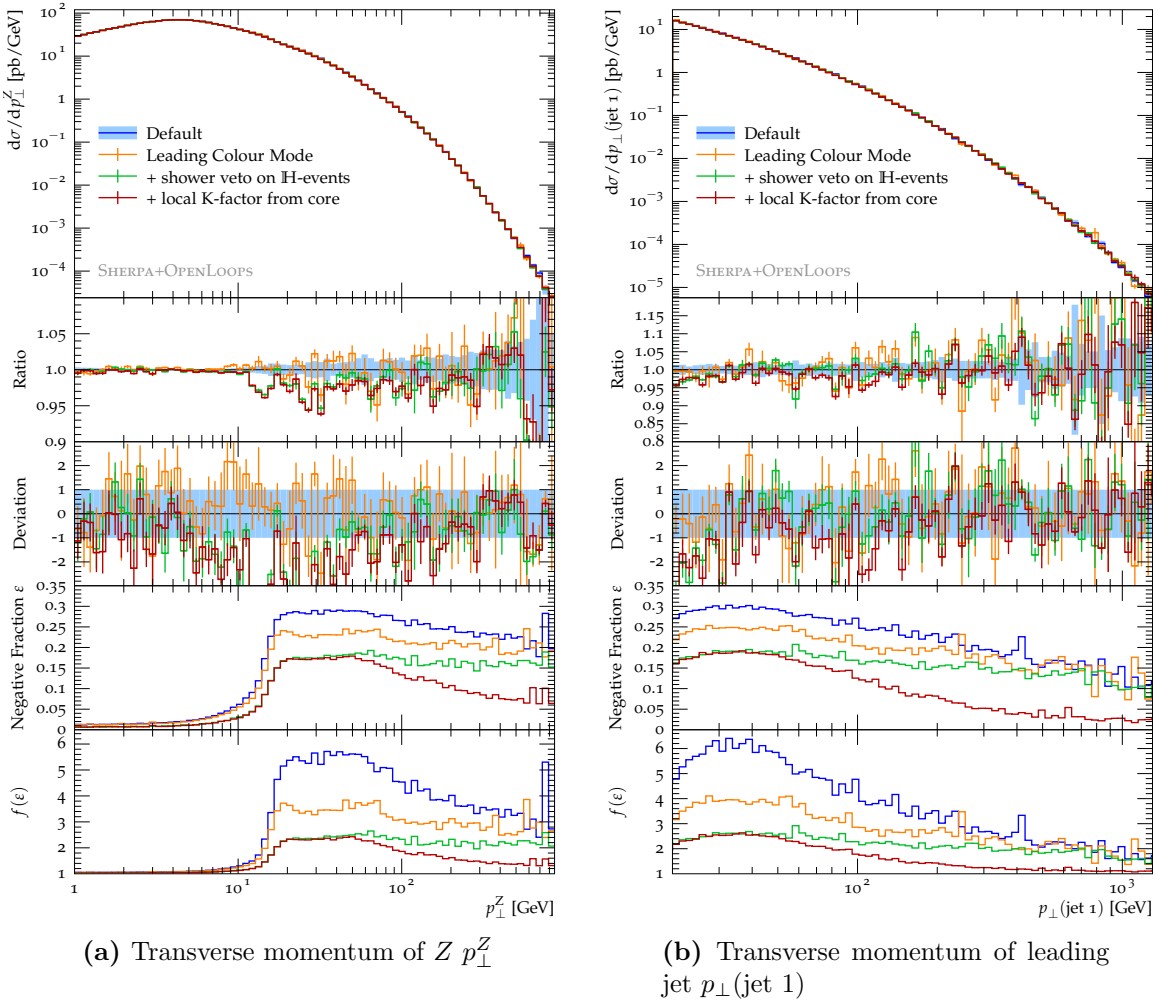

**(a)** Transverse momentum of $Z$ $p_{\perp}^{Z}$

**(b)** Transverse momentum of leading jet $p_{\perp}(\text{jet 1})$

**Figure 2:** *Monte Carlo validation observables comparing the different mechanisms to reduce the number of negative weighted events, as discussed in Section 4, for $pp \rightarrow Z+0,1,2$ jets@LO+3 jets@NLO. The lower two ratio plots display the negative weight fraction $\varepsilon$ and the corresponding factor $f(\varepsilon)$, which indicates how much more events need to be generated on average in order to achieve the same statistical accuracy as compared to only positive weighted events.*

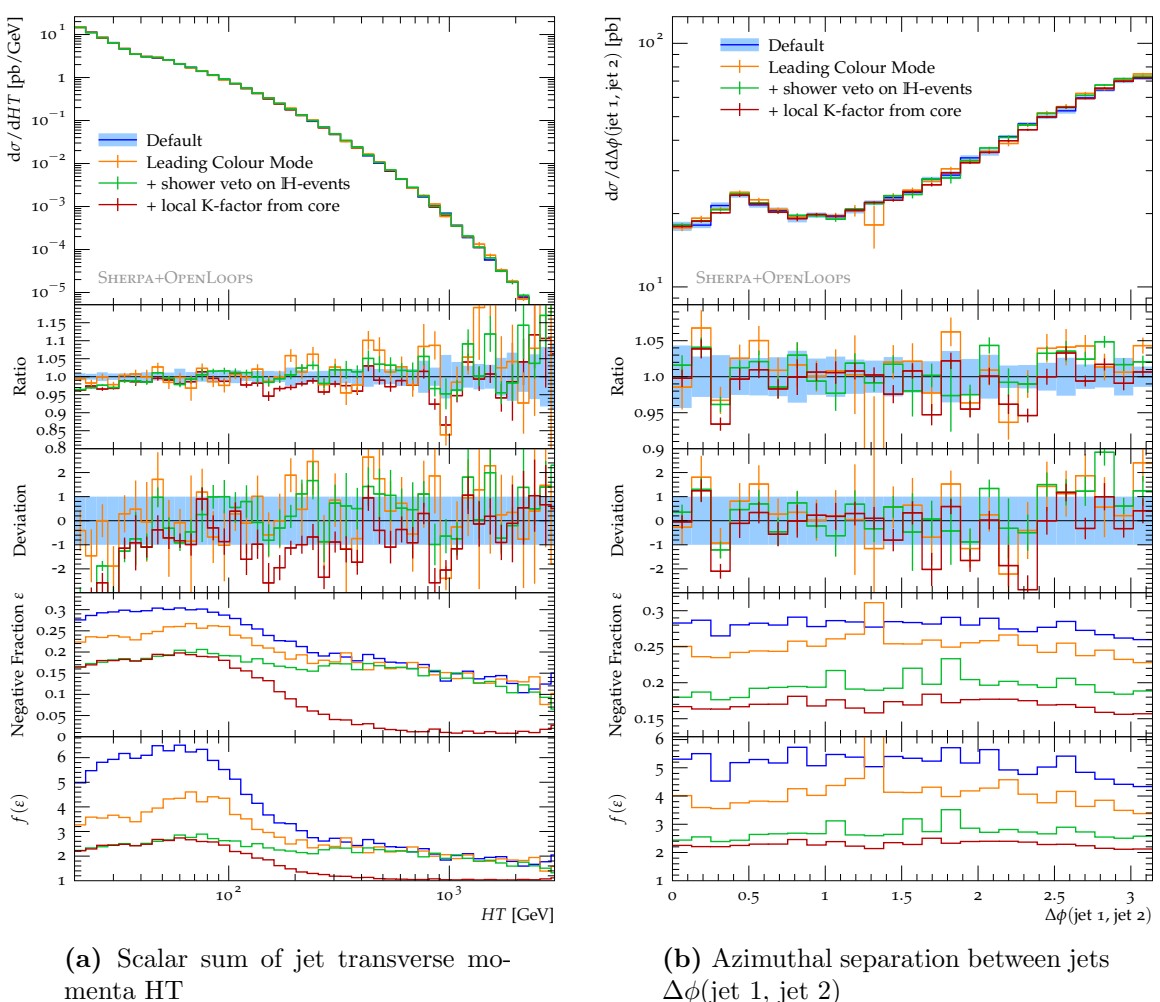

**(a)** Scalar sum of jet transverse momenta HT

**(b)** Azimuthal separation between jets $\Delta\phi(\text{jet 1, jet 2})$

**Figure 3:** *Monte Carlo validation observables comparing the different mechanisms to reduce the number of negative weighted events, as discussed in Section 4, for pp $\rightarrow$ Z+0,1,2 jets@LO+3 jets@NLO. See Fig. 2 for details.*

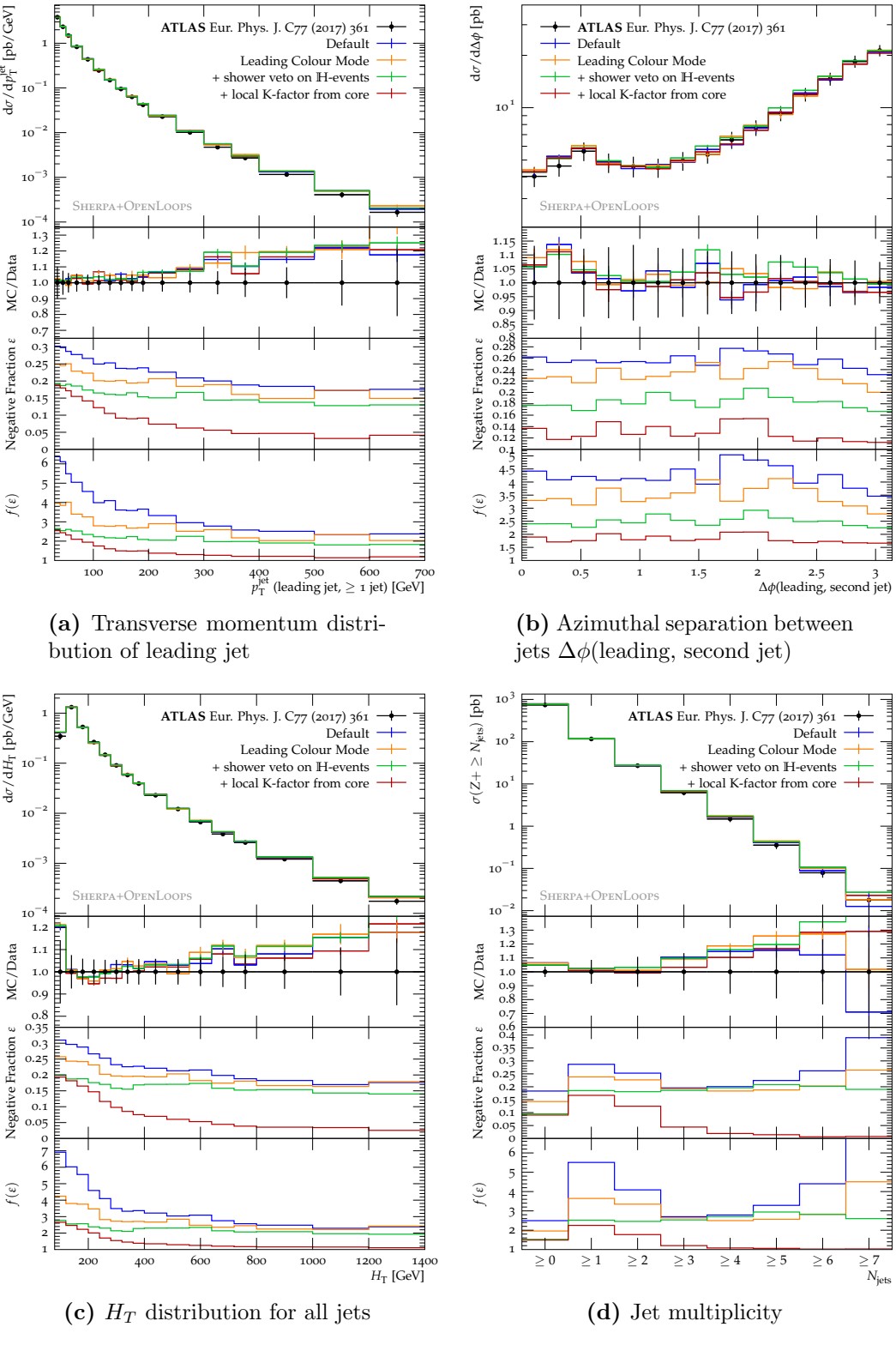

**(a)** Transverse momentum distribution of leading jet

**(b)** Azimuthal separation between jets $\Delta\phi$(leading, second jet)

**(c)** $H_T$ distribution for all jets

**(d)** Jet multiplicity

**Figure 4:** *Comparison of the different mechanisms as described in Section 4 to experimental Z+jets data measured by ATLAS [34]. See Fig. 2 for details.*

## 5.2  $t\bar{t}$+jets

The investigation of top quark pair production and decays in association with jets is crucial in the context of LHC physics. This process is one of the dominant backgrounds in almost all searches for new physics at the LHC, since it can also produce many jets in the final state and a high amount of missing transverse energy. For such large background samples it becomes particularly relevant to reduce the negative weight fraction.

|  | Negative Weight Fraction |
|---|---|
| Default | 24.8% |
| Leading Colour Mode | 18.7% |
| + shower veto on $\mathbb{H}$-events | 14.5% |
| + local K-factor from core | 12.6% |

**Table 3:**  *Evolution of the mean negative weight fraction for pp → $t\bar{t}$+0,1 jets@LO +2,3 jets@NLO.*

As can be seen from Table 3, regular $t\bar{t}$+jets samples contain approximately 25% of events with negative weights. This fraction can be approximately halved using the mechanisms as discussed in Section 4. In Figures 5 to 7, the different approaches aiming for the reduction of $\varepsilon$ are compared for various Monte Carlo validation observables.

Again, and despite now studying this non-trivial coloured core process, we find an excellent performance of the leading-colour approximation, which agrees with the nominal prediction in all observables. The same holds for the shower veto on $\mathbb{H}$ events, which does not lead to a deviation from the nominal prediction. A significantly stronger impact is observed in the setup where the local $k$-factor is calculated from the inclusive $t\bar{t}$ process. Especially distributions of the transverse momenta of jets are sensitive and yield a slope of up to 10% over the ranges studied here, resulting in significantly softer distributions. It is important to keep in mind, that especially these LO-accurate regions are plagued by large perturbative uncertainties [35, 36], significantly larger than the deviations found through the negative-weight reduction methods here.

A look at the differential impact on the negative weight fractions shows that the leading colour approximation reduces $\varepsilon$ in the soft region, and the shower veto on $\mathbb{H}$-events decreases the negatively weighted events in the soft to intermediate region even further. In the high $p_T$ region, $\varepsilon$ can be lowered only by defining the local K-factor using the lowest multiplicity process at NLO.

Figure 8 compares the Monte Carlo results to measurements of top quark pair production in association with jets for $pp$ collisions at a centre-of-mass energy of 13 TeV collected by the ATLAS detector, corresponding to an integrated luminosity of 3.2 fb$^{-1}$ [37]. In particular, the transverse momentum of hadronically decaying top quarks as well as of the $t\bar{t}$-system are investigated, where all tested methods are in fair agreement with the data.

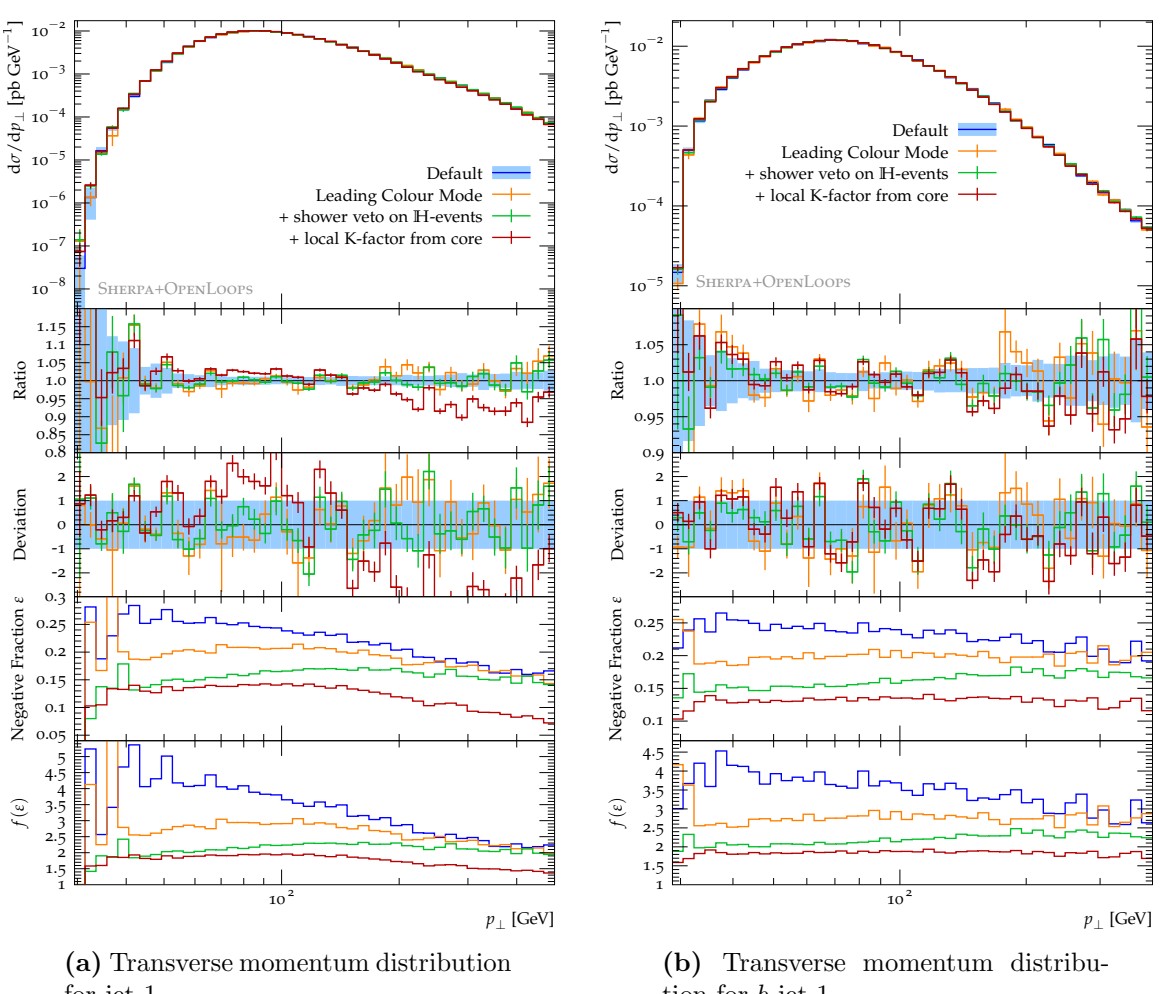

**(a)** Transverse momentum distribution for jet 1

**(b)** Transverse momentum distribution for $b$-jet 1

**Figure 5:** *Monte Carlo validation observables comparing the different mechanisms to reduce the number of negative weighted events, as discussed in Section 4, for pp $\rightarrow$ $t\bar{t}$+0,1 jets@LO+2,3 jets@NLO. See Fig. 2 for details.*

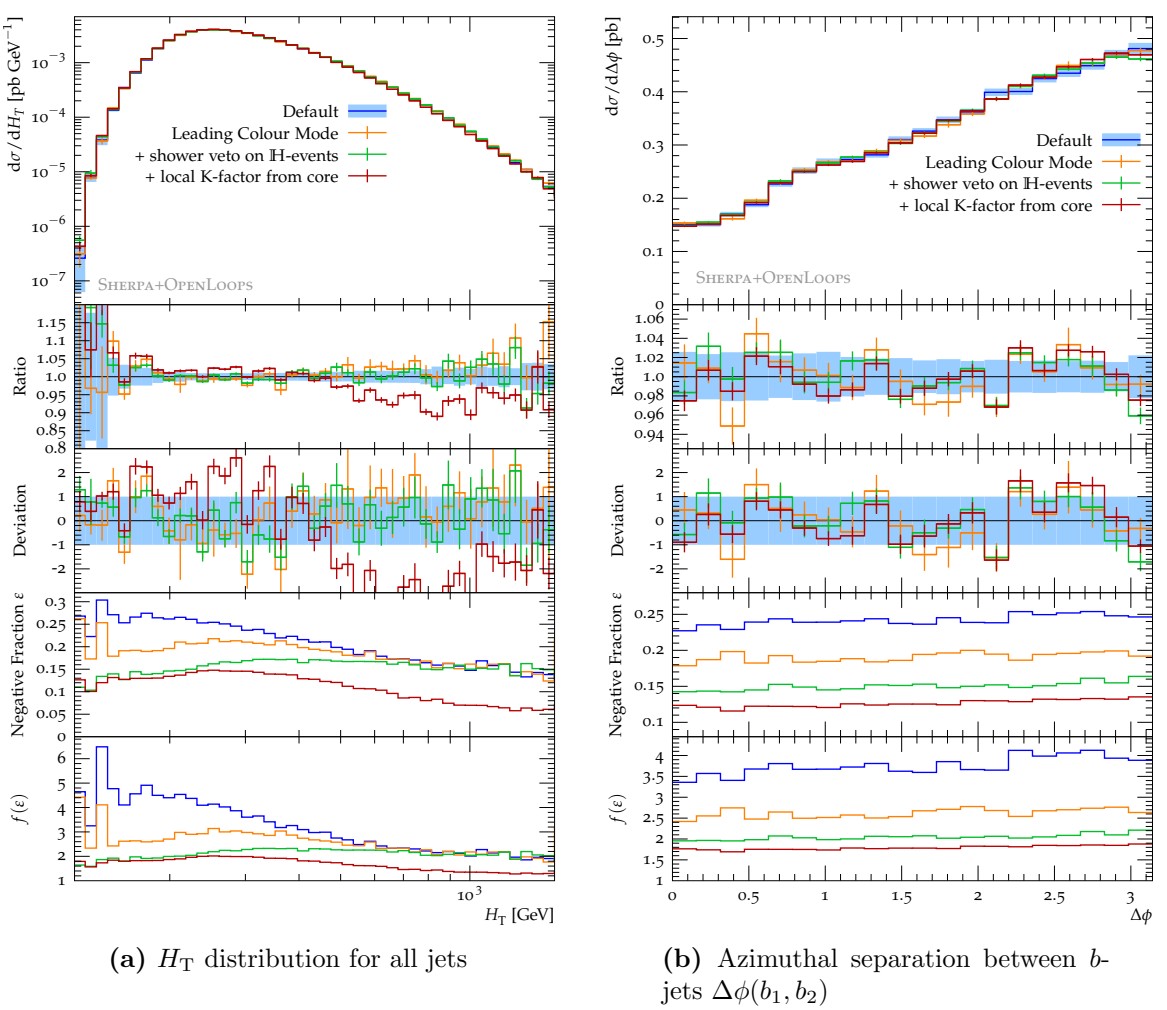

**(a)** $H_T$ distribution for all jets

**(b)** Azimuthal separation between $b$-jets $\Delta\phi(b_1, b_2)$

**Figure 6:** *Monte Carlo validation observables comparing the different mechanisms to reduce the number of negative weighted events, as discussed in Section 4, for $pp \rightarrow t\bar{t}+0,1$ jets@LO+2,3 jets@NLO. See Fig. 2 for details.*

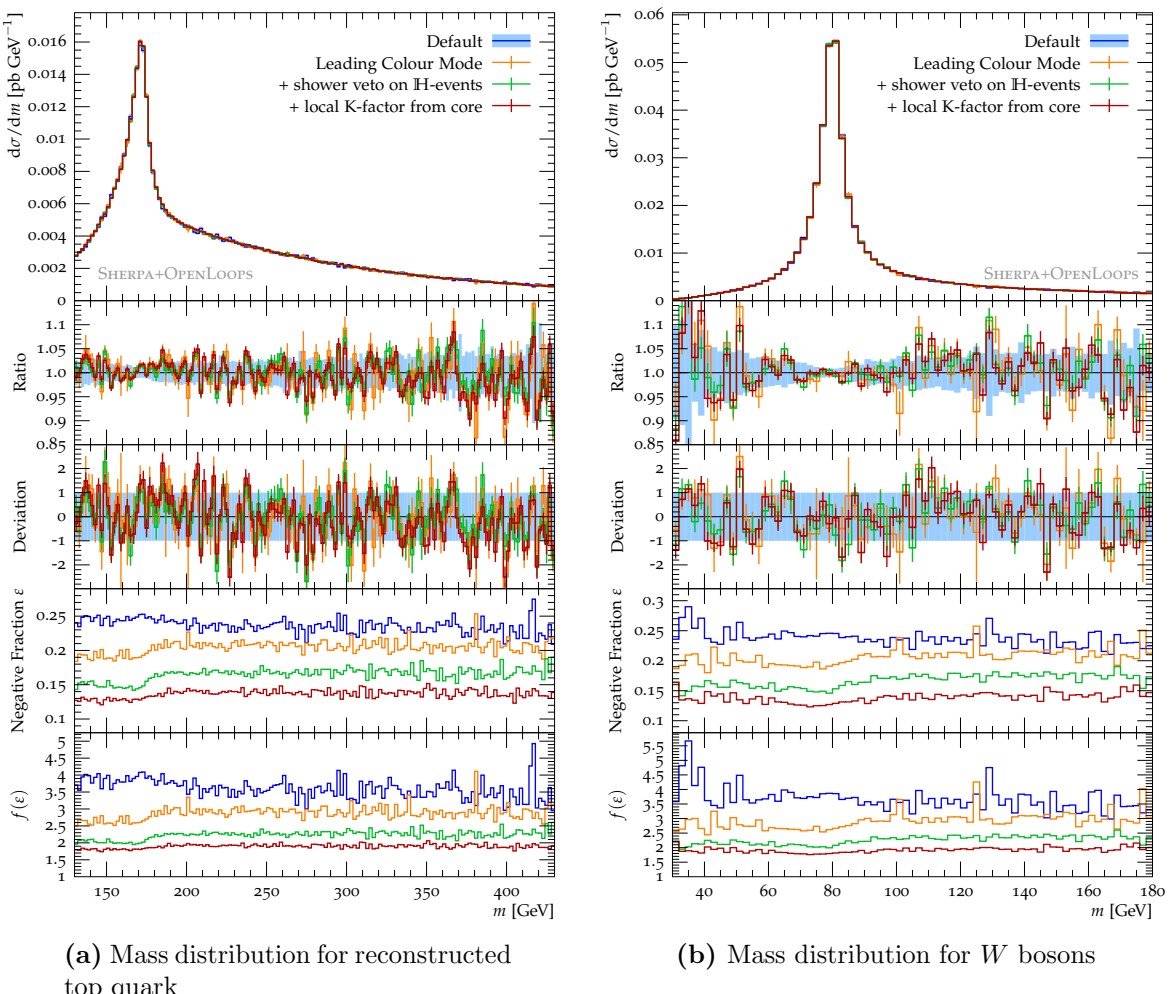

**(a)** Mass distribution for reconstructed top quark

**(b)** Mass distribution for $W$ bosons

**Figure 7:** *Monte Carlo validation observables comparing the different mechanisms to reduce the number of negative weighted events, as discussed in Section 4, for pp $\rightarrow$ $t\bar{t}$+0,1 jets@LO+2,3 jets@NLO. See Fig. 2 for details.*

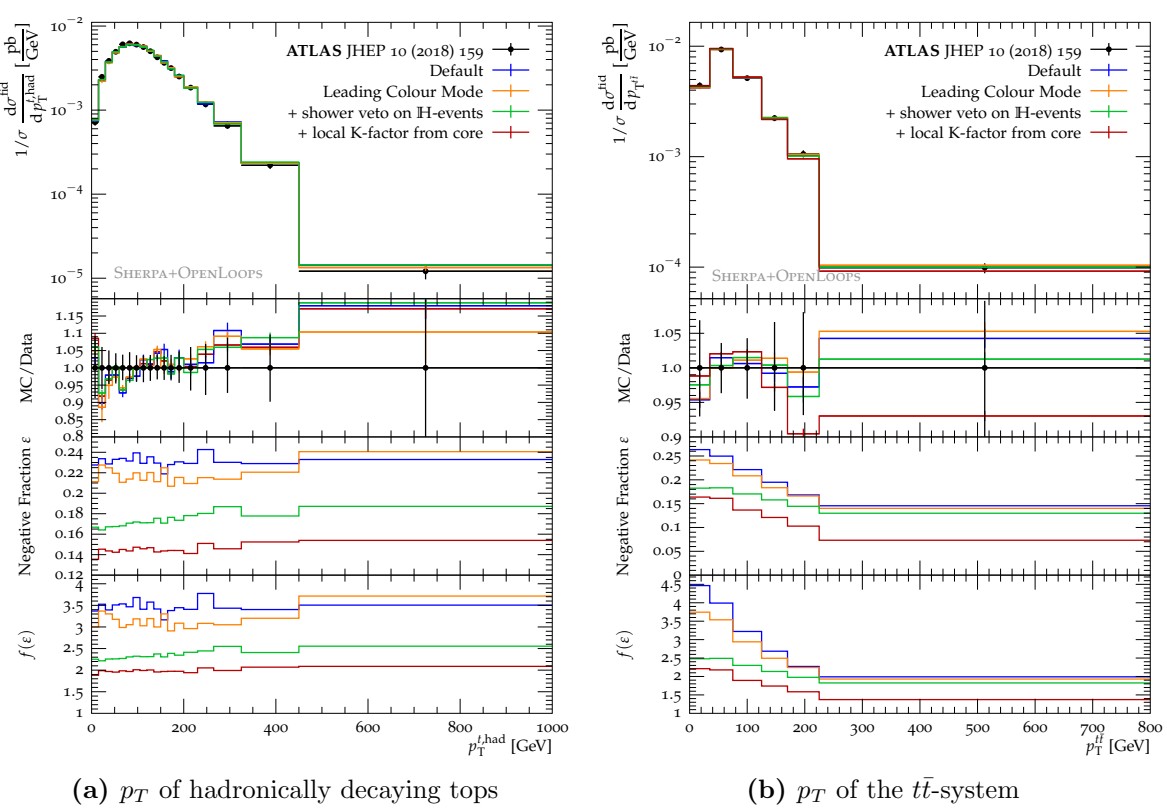

**(a)** $p_T$ of hadronically decaying tops

**(b)** $p_T$ of the $t\bar{t}$-system

**Figure 8:** *Comparison of the different mechanisms as described in Section 4 to experimental $t\bar{t}+jets$ data measured by ATLAS [37]. See Fig. 2 for details.*

# 6 Conclusions

We have presented methods to improve the computational footprint of SHERPA S-MC@NLO and MEPS@NLO simulations. The focus lies on a reduction of negatively weighted events, since these drastically reduce the statistical power of event samples of a given size and thus also impact upon the computational cost of the experimental detector simulation.

The sources of negative weights in the S-MC@NLO matching algorithm have been illustrated and their relative contributions quantified. Three mechanisms to reduce negative weights have been proposed. Some methods correspond to approximations which are valid within the claimed precision of the sample but can change the physics predictions of the simulation.

These proposals have been validated in the two most relevant processes, the production of a vector boson in association with jets and the production of a top-quark pair. The physics performance has been studied in typical observables to quantify the size of the changes induced. No critical differences were identified. At the same time, these methods were able to reduce the negative weight fraction by a factor of approximately two in the inclusive phase space. With the resulting negative weight fractions of $\varepsilon \approx 10\%$ the generation of large event samples becomes feasible again, which is particularly important in light of future high-luminosity LHC runs.

# Acknowledgments

We are grateful to our colleagues from the Sherpa collaboration for many fruitful discussions. FS and KD were supported by the German Research Foundation (DFG) under grant No. SI 2009/1-1. This research was supported by the Fermi National Accelerator Laboratory (Fermilab), a U.S. Department of Energy, Office of Science, HEP User Facility. Fermilab is managed by Fermi Research Alliance, LLC (FRA), acting under Contract No. DE-AC02-07CH11359. We thank the Center for Information Services and High Performance Computing (ZIH) at TU Dresden for generous allocations of computing time.

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
