# Peer review of "Reducing negative weights in Monte Carlo event generation with Sherpa"

_SciPost Physics_

## Round 1 · Referee Report · Anonymous · 2022-1-26

Strengths

1. Timely
2. Very clear presentation of an investigation into very relevant method

Weaknesses

1. While the approach reduces the number of negative weight, it fails to reproduce the results of the unmodified approach

Report

This report is very timely in presenting a study of three investigations into
reducing the occurrence of events with negative weight in the high-precision
fixed order and parton shower mergings possible in Sherpa. The presentation
is extremely clear and concise, and the paper is very well written.

The paper presents three unrelated approaches, and compares both the degree
to which the modifications reproduce the correct result, and the impact on
the frequency of negative weight events. One of the conclusions of the paper
is that (page 13) "In the high pT region, \epsilon can be lowered only by
defining the local K-factor using the lowest multiplicity process at NLO". I
have a question to the approach which would need to be investigated before
publication:

While it is correct that only the approach of the local K-factor has a
significant impact on the frequency of negative weight events, this approach
is also singled out by significant discrepancies in the distributions
obtained with both the unmodified approach and the other two approaches to
reducing the impact of negative weight events. For example, in figure 5a, the
very large pT region of interest, the method of the local K-factor
systematically undershoots the other approaches in all the last 16 bins, by
more than 3 standard deviations in several bins, and by more than two
standard deviations in 12 of the last bins. This surely is not just a
statistical fluctuation, but rather a systematic effect which would need
investigation. A similar systematic effect for the same method is seen in
figure 6. Systematic deviations are seen in the low HT-region in figure
3. However, these deviations are not discussed or compared with other
intrinsic uncertainty to the calculation. How do these differences compare to
even simple scale variations? There seems little point avoiding negative
weights at the cost of accuracy, since the negative weights arise only when
requiring higher accuracy on the predictions from the shower.

I am looking forward to seeing the result of this investigation.

Requested changes

1. Investigate whether the deviations discussed in the report are indeed systematic
2. Investigate the size of the deviations compared to the intrinsic uncertainty of the calculations

---

## Editorial Decision

editor-in-charge_assigned